# Weakly Supervised Semantic Parsing with Execution-based Spurious Program Filtering

**Kang-il Lee**[1]      **Segwang Kim**[2*]      **Kyomin Jung**[1,3†]

[1]Dept. of ECE, Seoul National University
[2]Samsung Electronics Mobile eXperience [3]IPAI, Seoul National University
{4bkang, ksk5693, kjung}@snu.ac.kr

## Abstract

The problem of *spurious programs* is a long-standing challenge when training a semantic parser from weak supervision. To eliminate such programs that have wrong semantics but correct denotation, existing methods focus on exploiting similarities between examples based on domain-specific knowledge. In this paper, we propose a domain-agnostic filtering mechanism based on program execution results. Specifically, for each program obtained through the search process, we first construct a representation that captures the program's semantics as execution results under various inputs. Then, we run a majority vote on these representations to identify and filter out programs with significantly different semantics from the other programs. In particular, our method is orthogonal to the program search process so that it can easily augment any of the existing weakly supervised semantic parsing frameworks. Empirical evaluations on the Natural Language Visual Reasoning and WIKITABLEQUESTIONS demonstrate that applying our method to the existing semantic parsers induces significantly improved performances. Code is available at https://github.com/klee972/exec-filter.

## 1 Introduction

Semantic parsing is the task of mapping natural language utterances into machine-executable meaning representations, often referred to as *programs*. Most deep learning-based semantic parsing studies take the supervised learning approach requiring utterance-program paired dataset. However, annotating such pairs demands expensive expert annotations. Instead, weakly-supervised semantic parsing, i.e. learning from denotation, has drawn much attention (Clarke et al., 2010; Liang et al., 2011; Berant et al., 2013). In this setup, a semantic parser is trained with cheaper denotation (execu-

tion result of the program) rather than the program itself.

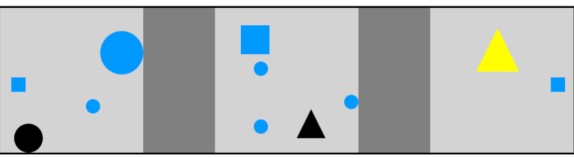

$x$ : There is a blue square
$w$ : [[{color: blue, shape: square}, {color: black, shape: circle}…], …]
$z$ : objExists(square(blue(all_objects)))
$z$' : objExists(black(circle(all_objects)))
$y$ : True

| Rank | Nation | Gold | Silver | Bronze | Total |
|------|--------|------|--------|--------|-------|
| 1 | Soviet Union | 50 | 27 | 22 | 99 |
| 2 | United States | 33 | 31 | 30 | 94 |
| 3 | East Germany (GDR) | 20 | 23 | 23 | 66 |
| 4 | West Germany (FRG) | 13 | 11 | 16 | 40 |
| 5 | Japan | 13 | 8 | 8 | 29 |
| 6 | Australia | 8 | 7 | 2 | 17 |
| 7 | Poland | 7 | 5 | 9 | 21 |
| 8 | Hungary | 6 | 13 | 16 | 35 |
| 9 | Bulgaria | 6 | 10 | 5 | 21 |
| 10 | Italy | 5 | 3 | 10 | 18 |

$x$ : How many nations won more than ten silver medals?
$w$ : [[{Rank: 1}, {Nation: Soviet Union}, {Gold: 50}…], …]
$z$ : count(filterNumberGreater(allRows, column:Silver, 10))
$z$' : select(filterIn(allRows, column:Nation, Japan), column:Rank)
$y$ : 5

Figure 1: Overview of task setup on Natural Language Visual Reasoning (top) and WIKITABLEQUESTIONS (bottom) dataset. The datasets include only utterance $x$, world $w$ and denotation $y$ (ground truth program $z$ is not given). Spurious programs like $z'$, whose meaning is wrong but execution result is correct, are major challenges of the task.

Without supervision for correct programs, training a weakly supervised semantic parser typically entails a program search process. In this process, given the natural language utterance, a search algorithm such as beam search generates a pool of likely programs. Among these programs, there may be some programs that have incorrect semantics but derive the correct denotation by chance, as $z'$ in Figure 1. These programs are called *spurious programs* and introduce undesirable noise on the training signal. Hence, filtering these spurious programs is of great interest in weakly supervised

---

*Work done while at Seoul National University.
†Corresponding author.

semantic parsing (Pasupat and Liang, 2016; Goldman et al., 2018).

Prior works on weakly supervised semantic parsing attack the spuriousness problem by introducing some domain-specific knowledge, such as abstracted utterance-program pairs (Goldman et al., 2018) or utterance groups (Gupta et al., 2021). Unlike these works, we propose a domain-agnostic filtering mechanism based on a majority vote over program execution results to alleviate the spuriousness issue. Our intuition is that programs whose execution results largely deviate from those of other programs in the pool are likely to be spurious, thus filtering them out would improve the training of weakly supervised semantic parsers. To effectively quantify the degree of deviation, we propose a novel representation scheme of programs based on the execution results. Here, the entries of a representation vector are the program's execution results on worlds retrieved from other training examples. To exclude spurious programs, we run a majority vote on these representations to construct a "centroid representation" and filter out the programs whose representation is dissimilar to it. Our method can be applied to any weakly supervised semantic parser with minimal modification, as long as it involves a program search step in the training process.

We evaluate our filtering mechanism on two challenging datasets with distinct characteristics: Natural Language Visual Reasoning (NLVR) (Suhr et al., 2017) and WIKITABLEQUESTIONS (WTQ) (Pasupat and Liang, 2015). When added on the base models (Gupta et al., 2021; Wang et al., 2019), our filtering mechanism shows significant improvement over the baselines without using additional domain-specific knowledge. Finally, we quantitatively analyze the effectiveness of our approach in detecting spurious programs and conduct an error analysis on a failure case.

## 2 Background

In this section, we formalize weakly supervised semantic parsing problems and introduce two datasets: NLVR and WTQ.

### 2.1 Problem Definition

The dataset for weakly supervised semantic parsing consists of $N$ examples $\{x_i, w_i, y_i\}_{i=1}^N$, where $x_i$ is a natural language utterance, $w_i$ is a set of worlds that $x_i$ can be evaluated on, and $y_i$ is a set of deno-

tations indicating the semantic of $x_i$ in each world. Our goal is to train a model such that when given $x_i$ as input, it produces a program $z_i$, which returns (each member of) $y_i$ when executed on (each member of) $w_i$.

### 2.2 Datasets

**NLVR** Natural Language Visual Reasoning (Suhr et al., 2017) is a dataset of blocks world domain that requires complex reasoning abilities. The world is given structured representations of various objects and the utterance is a statement about the properties or relations of the objects in the world, as shown in Figure 1 (Here, we graphically display the world to help the reader understand). The denotations are Boolean values representing whether the given utterance is true or false in the world. There are 3,163 unique training examples consisting of one utterance and four world-denotation pairs. Also, there are development, public test, and hidden test sets with 267, 266, and 266 examples each.[1]

**WTQ** WIKITABLEQUESTIONS (Pasupat and Liang, 2015) is a table semantic parsing dataset with complex queries and large natural language variation. Worlds are structured representations of Wikipedia tables and utterances are questions about the tables. Unlike NLVR, denotations can have values in the table cells or values obtained by applying some elementary functions to the cell values. The WTQ training set consists of 11,321 training examples and 2,831 development examples. It provides 4,344 test examples with unseen tables to measure the model's generalization performance.

## 3 Execution-based Filtering

To eliminate spurious programs, we devise a novel execution-based filtering mechanism. Intuitively, among consistent programs, the spurious ones are likely to semantically deviate. However, measuring or defining semantic distance is challenging. Thus, we instead loosely capture the semantics of programs by executing them against reasonably selected worlds. Then, we filter out the programs whose execution results deviate most from others by performing a majority vote.

**Formal Setups** Consider an example with utterance $x$, world $w$, and denotation $y$.[2] The programs

---

[1]The hidden test set is now made public by dataset creators.
[2]We omit the data index for brevity.

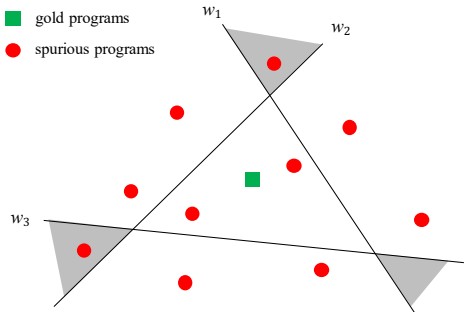

Figure 2: Illustration of our program representation scheme and filtering based on majority vote. Retrieved worlds ($w_j$'s) partition the programs into several groups by their execution results and are represented as lines in the figure.[4] By running majority vote based on the execution results, programs in the gray regions may be filtered.

found in the search step are executed against $w$, and only those with correct denotations remain in the program pool $\{z_i\}_{i=1}^k$.[3] Still, many of these $k$ programs may be spurious; they do not reflect the meaning of given utterance $x$ but coincidentally derive correct denotation $y$.

### 3.1 Program Representation

In order to capture the semantics of programs, we devise a representation scheme based on their execution results against a set of worlds. Mathematically speaking, we represent a semantic of program $z_i$ as an $n$-dimensional sparse vector $r_i$ whose $j$-th entry is the execution result of $z_i$ against world $w_j$. Regarding the worlds $\{w_j\}_{j=1}^n$, we collect them from the training set using two different selection strategies for NLVR and WTQ, on which we elaborate in section 4 and appendix B.

### 3.2 Filtering Programs with Majority Vote

The program representations $\{r_i\}_{i=1}^k$ can be understood as points on a space and are partitioned by $\{w_j\}_{j=1}^n$. As shown in Figure 2, $r_i$ can be classified based on which region it resides. We hypothesize that the programs far from the "centroid" of $\{r_i\}_{i=1}^k$ are more likely to be spurious. Here, we employ two vote techniques: (1) a "hard" vote that constructs explicit centroid representation utilizing only the winning denotations and (2) a "soft" vote that considers the proportion of each denotation.

---

[3]In NLVR, one utterance typically has four worlds and denotations. Therefore, a program remains in the pool when it correctly produces all four denotations.

[4]In this illustration, we assume binary denotations so that a world partitions programs into two groups.

**Hard Vote**  In order to get centroid program representation $r_*$ with hard vote, we run majority vote along each entry of $\{r_i\}_{i=1}^k$:

$$r_*^j = \underset{e \in E}{\operatorname{argmax}} \sum_{i=1}^k \mathbb{1}(r_i^j = e) \qquad (1)$$

where $r^j$ is the $j$-th entry of vector $r$ and $\mathbb{1}$ is an indicator variable. $E$ denotes the set of denotations obtained by executing programs in the pool. Note that $r_*$ is not necessarily the same as the representation of the real gold program. Given the representation $r_i$, we use $r_*$ to define the score $s_i$ as follows:

$$s_i = \frac{1}{n} \sum_{j=1}^n \mathbb{1}(r_i^j = r_*^j). \qquad (2)$$

Finally, we filter out the programs with a score lower than the heuristically chosen threshold $\tau$.

We can improve this mechanism by weighting representations when performing the majority vote, proportional to some scalar metric of the goodness of the program. This metric can be either defined manually with domain-specific knowledge or learned from data. We add this weight term $W(z)$ to equation 1 as follows:

$$r_*^j = \underset{e \in E}{\operatorname{argmax}} \sum_{i=1}^k W(z_i)\mathbb{1}(r_i^j = e). \qquad (3)$$

As metric $W(\cdot)$, one can use a program's model likelihood or hand-crafted scores such as lexicon coverage used by Dasigi et al. (2019) and Gupta et al. (2021). Note that this weighting technique is different from soft vote, which we describe next.

**Soft Vote**  One drawback of hard vote is that it cannot take into account the proportion of denotations because only the eventual winner is represented as centroid representation. Instead, we can incorporate the denotation proportion into the score $s_i$ by counting the number of programs with the same execution result as $z_i$.

$$s_i = \sum_{j=1}^n \sum_{l=1}^k \mathbb{1}(r_i^j = r_l^j) \qquad (4)$$

As in hard vote, each program's contribution can be weighted by $W(\cdot)$:

$$s_i = \sum_{j=1}^n \sum_{l=1}^k W(z_l)\mathbb{1}(r_i^j = r_l^j). \qquad (5)$$

We normalize the scores $\{s_i\}_{i=1}^k$ such that the highest score becomes 1 and filter out the programs whose normalized score is lower than some threshold $\tau$.

## 4  Collecting Execution Results

In the previous section, the program representation is defined by the program's execution results on multiple worlds. Now we describe methods to select the worlds and obtain programs' execution results on those worlds. We discuss two separate approaches for NLVR and WTQ, considering their vastly different world configurations.

### 4.1  Collecting Execution Results for NLVR

In NLVR, a program representation $r_i$ consists of Boolean execution results of $z_i$ on various worlds $\{w_j\}_{j=1}^n$. In order to effectively identify outliers among the programs, the representations $\{r_i\}_{i=1}^k$ should provide helpful information when performing the majority vote. For example, a world $w_j$ that returns 'True' for all $z_i$'s is useless because we would not get any information to identify outliers from the $j$-th entry. That is, worlds used for building representations should be able to classify $z_i$'s according to their execution results. Our hypothesis is that the worlds whose corresponding utterance is similar to $x$ would be more informative than others because these worlds are more likely to be relevant to the meaning of $x$. To this end, we collect worlds from the training set by retrieving top-$n$ ($n = 80$ in NLVR) worlds $\{w_j\}_{j=1}^n$ based on the BLEU score between $x$ and each $w_j$'s corresponding utterances.

### 4.2  Collecting Execution Results for WTQ

As each program in WTQ is conditioned on a specific table (denoted as *source table* henceforth) and therefore cannot be used on others, we propose a method for modifying programs so that they can be executed on a *target table* we want. In particular, we replace entity and column names in the programs with those in the target table to make the programs executable while maintaining the semantic relationship between the programs.

Consider an example with a program pool $Z = \{z_i\}_{i=1}^k$ and a target table $w$. Our goal is to replace all the column and entity names in $Z$ with those in $w$ while maintaining the semantic relationships between the programs in $Z$. Thus, we replace the names *consistently*, i.e., the constants of identical name in multiple programs should be replaced with

Figure 3: Illustration of column and entity replacement. Here, $z_1$ and $z_2$ are programs conditioned on the source table, and $z_1'$ and $z_2'$ are their counterparts modified to be executed on the target table. Within the programs, column and entity names of the same type are displayed in the same color.

a single name in the target table. Specifically, we first find all the column and entity names in $Z$ and $w$ and identify their types, e.g., string, number, etc. Then, for each name $N$ in $Z$, we randomly sample a name $N'$ with the same type in $w$ and replace all the occurrences of $N$ with $N'$. We illustrate the column and entity replacement in Figure 3, where programs $z_1$ and $z_2$ are modified to $z_1'$ and $z_2'$. Note that all the occurrences of column:Wins are replaced with column:Silver consistently.

During the execution of modified programs on a target table, some return execution errors for various reasons. For example, the function argmax in the domain language used by Wang et al. (2019) returns an error when the input is a list of length one. When there are too many errors (when more than 10% of programs return error in our experiments) for particular table, we perform the sampling and replacement process again. The resampling procedure may be repeated up to 10 times per table, and we discard the table if no satisfactory replacement is found after 10 iterations. The program representation is constructed by repeating this process until the number of worlds used hits $n$ ($n = 40$ in WTQ). More details on the table collection is described in appendix B.

## 5  Experiments

To validate the efficacy of our filtering method, we conduct experiments on NLVR and WTQ datasets. First, for each dataset, we characterize existing parsers we use and explain where our method is applied. After explaining our filtering implementation on each of these datasets, we present our

experimental results and ablation studies.

## 5.1 NLVR Implementation Details

**NLVR Base Parser** We use the Iterative Search (Dasigi et al., 2019) and Consistency-based Parser (Gupta et al., 2021) to construct a base parser. In this section, we provide a background on these two methods.

Iterative Search (Dasigi et al., 2019) uses grammar-constrained RNN encoder-decoder model of Krishnamurthy et al. (2017). Its training undergoes alternating two steps: Maximum Marginal Likelihood (MML) and Minimum Bayes Risk (MBR). In MML, when given the utterance $x_i$ and denotation $y_i$, the parser is trained to maximize the likelihood of marginalization of programs $z_i$'s that are consistent with $y_i$. In MBR, the model is initialized with the previous MML checkpoint and trained to minimize cost functions for denotation accuracy and lexicon coverage.

Consistency-based Parser (Gupta et al., 2021) improves the consistency of Iterative Search by introducing Logical Language Design (LLD) and Consistency Reward (CR). In LLD, they replace some macro functions with generic functions that are more reusable across different contexts. Also, they introduce the Consistency Reward (CR), which encourages consistency between programs for related utterances.

In both approaches, there is a program search step between every MBR and MML to construct a dataset to train the model in the subsequent MML. The model performs a beam search in this search step to produce a pool of likely programs. Our filtering mechanism is attached here to minimize undesirable noise from spurious programs being included in the next step's MML dataset.

**Lexicon-based Program Search** Our initial implementation without modifying the base model exhibited a decrease in performance, presumably due to the beam size not being suitable for our method. As our filtering mechanism relies on the majority vote, it would be advantageous if a large and informative candidate program pool is obtained during the search step. To this end, instead of simply enlarging the beam size, we apply the more efficient beam search augmented by the lexicon, which is utilized by Dasigi et al. (2019) and Gupta et al. (2021).

With the utterance $x$, its lexicon was given a set of program tokens $\mathcal{A}(x)$ that should be in program

$z$. We defined a lexicon recall score $R(\mathcal{A}(x), z)$ which is the number of tokens included in both $\mathcal{A}(x)$ and $z$, divided by $|\mathcal{A}(x)|$. When performing a beam search, we doubled the beam size and performed the beam search again until the program pool $\{z_i\}_{i=1}^{k}$ had at least one program $z_i$ with $R(\mathcal{A}(x), z_i) = 1$. Doubling the beam size was stopped if a program with a score of 1 was not found after 5 iterations. We used lexicon recall score $R(\mathcal{A}(x), z)$ as a proxy to the example difficulty (and the number of consistent programs in the beam) because when the utterance is complex, the model usually produces a very small number of consistent programs with low lexicon recall scores. We empirically observed that there is no computational issue with such a large beam size because most of the nodes in the beam reach dead ends due to the grammar constraint.

For the filtering mechanism, we used this lexicon recall score $R(\mathcal{A}(x), z)$ as the weight term $W(\cdot)$ described in section 3.2. We used hard vote on NLVR and chose the filtering threshold $\tau$ as 0.8.

## 5.2 WTQ Implementation Details

**WTQ Base Parser** As a base parser for WTQ, we use the Structured Attention (Wang et al., 2019), which is a strong and light-weighted system suitable for testing our filtering mechanism. Among the systems that do not utilize pre-training on external datasets, it is state-of-the-art on WTQ.

Unlike Iterative Search and Consistency-based Parser, they exhaustively search the programs consistent with the given denotation prior to the training. The exhaustive search is manageable because the space of programs is significantly reduced by introducing abstract programs and restricting the number of production rules. In this case, we take a fine-tuning approach: the model is first trained as proposed by Wang et al. (2019), and then fine-tuned with programs filtered with our method. The model was trained using the official code released by the authors, achieving slightly worse results (dev accuracy of 43.2 and test accuracy of 44.4) than those reported in the paper. Then, we fine-tuned this reproduced model for 5 epochs with programs filtered with our method.

We used the instantiation model likelihood of the base parser (Wang et al., 2019) as weight terms $W(\cdot)$ representing how well the program aligns with the given utterance. Besides, we employed soft vote with filtering threshold $\tau = 0.2$. More

| | Dev. | | Test-P | | Test-H | | Test |
|---|---|---|---|---|---|---|---|
| **Approach** | **Acc.** | **Con.** | **Acc.** | **Con.** | **Acc.** | **Con.** | **Con.** |
| Abs. Sup. + ReRank (Goldman et al., 2018) | 85.7 | 67.4 | 84.0 | 65.0 | 82.5 | 63.9 | 64.5 |
| Iterative Search (Dasigi et al., 2019) | 85.4 | 64.8 | 82.4 | 61.3 | 82.9 | 64.3 | 62.8 |
| LLD (Gupta et al., 2021) | 88.2 | 73.6 | 86.0 | 69.6 | 87.2 | 70.1 | 69.9 |
| LLD + CR (Gupta et al., 2021) | 89.6 | 75.9 | 86.3 | 71.0 | 89.5 | 74.0 | 72.5 |
| LLD (w/ modified beam search) | 90.8 | 77.8 | 88.3 | 73.4 | 89.0 | 74.6 | 74.0 |
| + Execution-based Filtering | 90.5 | **78.8** | **89.4** | 74.2 | **89.4** | **76.3** | **75.2** |
| LLD + CR (w/ modified beam search) | 90.3 | 77.5 | 87.8 | 72.8 | 87.8 | 72.2 | 72.5 |
| + Execution-based Filtering | **90.9** | 78.7 | 88.7 | **74.9** | 88.8 | 72.5 | 73.7 |

Table 1: Accuracy and consistency of our approach and prior works on NLVR development, test-public (Test-P), and test-hidden (Test-H) sets. The rightmost column (Test) shows the average consistency of Test-P and Test-H. LLD: Logical Language Design. CR: Consistency Reward.

implementation details for NLVR and WTQ are described in appendix A.

## 5.3 Main Results

In both NLVR and WTQ, we evaluate the models with *accuracy*, which considers the correctness of the execution result for only one world-denotation pair. For NLVR, we also use *consistency*, which counts a program as consistent if it is correct in all four world-denotation pairs. We report the average value of 4 runs with different random seeds.

**NLVR**  Our modification on beam search described in section 5.1 yields quite different results depending on the setting; there is a significant improvement when the LLD is used alone, but a rather small change in LLD + CR setup as shown in the Table 1. Interestingly, the consistency reward is not helpful in our modified versions. When our filtering mechanism is applied to these modified base models, it improves the test accuracy and consistency in both of base models as shown in Table 1. When we add our filtering mechanism on our modified version of LLD, the average test consistency improves by 1.2%. Adding our mechanism on the modified LLD + CR also shows an improvement of 1.2% in test consistency. We follow Gupta et al. (2021) and perform a statistical test on the significance of the improvements with Deep Dominance (Dror et al., 2019). All improvements mentioned turn out to be statistically significant (p < 0.05), implying that our approach effectively filters out spurious programs.

**WTQ**  As shown in table 2, our filtering mechanism improves the base parser (Wang et al., 2019)

| **Approach** | **Dev.** | **Test** |
|---|---|---|
| Zhang et al. (2017) | 40.4 | 43.7 |
| Liang et al. (2018) | 42.3 | 43.1 |
| Dasigi et al. (2019) | 42.1 | 43.9 |
| Agarwal et al. (2019) | 43.2 | 44.1 |
| Wang et al. (2019) | **43.7** | 44.5 |
| + Execution-based Filtering | 43.2 | **44.8** |

Table 2: Accuracy of our approach and previous works on WIKITABLEQUESTIONS development and test sets.

| | **NLVR** | | **WTQ** | |
|---|---|---|---|---|
| **Vote Type** | **Dev.** | **Test** | **Dev.** | **Test** |
| Hard Vote | **78.8** | **75.2** | 42.8 | 44.3 |
| Soft Vote | 77.0 | 73.4 | **43.2** | **44.8** |

Table 3: Consistency (NLVR) and accuracy (WTQ) with hard vote and soft vote.

in the test set, which consists of the tables unseen during the training. This result demonstrates that our method successfully filters out spurious programs and thus the parser learns more generalizable regularities in the mapping of natural language to the program. Our approach achieves the highest reported test accuracy among semantic parsers that do not make use of any external data.

## 5.4 Impact of Vote Types

As mentioned in section 5.1 and 5.2, we use different vote types for NLVR and WTQ. Empirical result in table 3 shows a distinct superiority of the vote methods in each domain. In NLVR, hard vote

| Weight Type | Dev. | Test |
|---|---|---|
| Lexicon Recall Score | **78.8** | **75.2** |
| Model Likelihood | 77.0 | 72.5 |
| w/o Weight Term | **78.8** | 74.1 |

Table 4: Consistency on NLVR with different weight types.

| $\tau$ | Precision | Recall | F1-score |
|---|---|---|---|
| 0.8 | 99.5 | 40.0 | 49.5 |
| 0.9 | 99.6 | 57.8 | 66.3 |
| 1.0 | 99.4 | 82.0 | 85.7 |

Table 5: Spurious program detection performance on 30 NLVR training examples at the last search step, with various threshold $\tau$. All the values are calculated individually for each train example and averaged afterward.

outperforms soft vote, but the opposite is true in WTQ. This tendency can be explained by the distinct characteristics of two domains. NLVR has Boolean denotation, thus the winning denotation always gets more than 50% of the vote and can represent the programs' semantics. However, in WTQ, the denotations can have various values, making the winner much less representative of the entire program pool compared to that in NLVR. Soft vote can alleviate this issue by considering the denotations of all programs in the pool.

### 5.5 Impact of Weight Term $W(\cdot)$

The choice of weight term $W(\cdot)$ is another important factor in our filtering mechanism. In this section, we analyze the effectiveness of three different types of vote weighting in NLVR: (1) lexicon recall score $R(\mathcal{A}(x), z)$, (2) model likelihood $p(z|x)$, and (3) vote without weight term (which corresponds to the equation 1).

As shown in table 4, the use of lexicon recall score exhibits the most improvement over the base parser, indicating the effectiveness of weighting votes with a metric of alignment between the utterance and program. Vote without any weight term shows weaker performance improvement. Interestingly, weighting with model likelihood deteriorates the performance, presumably due to positive feedback of up-weighting spurious programs through the training process.

## 6 Analysis

**Setup** In this section, we quantitatively analyze the effectiveness of our approach in filtering out spurious programs. To assess its effectiveness in distinguishing between spurious and correct programs, we randomly select 30 examples from the NLVR training set and manually label the programs obtained through the last program search step. Next, we evaluate the performance of spurious program detection, by classifying all programs with scores lower than the threshold $\tau$ as spurious.

**Spurious Program Detection** In table 5, we report the precision, recall, and F1-score in spurious program detection for various thresholds $\tau$. High precision values show that the majority of semantically correct programs have scores very close to $1.0$. This suggests that the centroid representation obtained through majority vote closely approximates the true gold program representation.

We also discovered that the optimal threshold for detecting spurious programs does not align with the optimal threshold for NLVR task performance. In our main experiment, the best NLVR test consistency is attained when using a threshold value of $\tau = 0.8$. However, it appears that this threshold is somewhat generous, as 60% of spurious programs remain in the program pool. When we raised the threshold to $\tau = 1.0$, which is the optimal value for detecting spurious programs, we noticed a decrease in NLVR test consistency. One potential reason for this phenomenon is that a high threshold produces too many false positives in the early stages of training, which hampers the search space exploration throughout the training process. To address this trade-off, one possible direction for future work is to utilize an adaptive threshold for each step of the search process.

**Correlation Statistics** To further analyze our method in depth, we report some correlation statistics between the programs' score and their spuriousness. First, the Pearson correlation coefficient between the spuriousness label (1 if spurious, 0 if not spurious) and $1 - s_i$, where $s_i$ is a program score described in equation 2, is $0.358$. Also, the ROC-AUC score stands at $0.738$ when using the score $s_i$ to classify whether the program is spurious or not. Finally, the mean and standard deviation of scores $s_i$ for correct programs are $0.997$ and $0.029$ respectively, while for spurious programs, they are $0.899$ and $0.155$.

**(Successful case) Sentence: There is at least one black item closely touching the bottom of a box.**

| Score | Program |
|---|---|
| 1.0 | **((* (* (object_count_greater_equals 1) black) touch_bottom) all_objects)** |
| 1.0 | **((* (* object_exists black) touch_bottom) all_objects)** |
| 0.85 | ((* (* (* (object_count_greater_equals 1) black) touch_bottom) bottom) all_objects) |
| 0.58 | ((* (* (object_count_greater_equals 2) black) touch_bottom) all_objects) |
| 0.50 | (box_count_greater_equals 2 (box_filter all_boxes (* (* (object_count_greater_equals 1) black) touch_bottom))) |

**(Failure case) Sentence: There are 2 black blocks**

| Score | Program |
|---|---|
| 1.0 | ((* (object_count_greater_equals 2) black) all_objects) |
| 1.0 | (object_exists (object_in_box all_boxes)) |
| 0.63 | (box_count_equals 2 (box_filter all_boxes (* object_exists black))) |
| 0.62 | **((* (object_count_equals 2) black) (object_in_box all_boxes))** |
| 0.62 | **((* (object_count_equals 2) black) all_objects)** |

Table 6: Successful (top) and failure (bottom) case of our filtering mechanism. Boldfaced programs are semantically correct programs and the others are spurious programs.

**Successful and Failure Cases** Table 6 shows both successful and failure cases in the last search step on NLVR train set. In the successful case, all programs with score lower than 0.8 are filtered out and turn out to be all spurious. However, a failure occurred when the majority vote selected false program between two programs with semantically similar functions (object_count_equals and object_count_greater_equals).

## 7 Related Work

### 7.1 Weakly Supervised Semantic Parsing

Recent research on semantic parsing has focused on *weakly-supervised semantic parsing*, or *learning from denotations*, whose goal is to learn a semantic parser without manual program annotation (Clarke et al., 2010; Liang et al., 2011; Berant et al., 2013). In an effort to reduce the program search space and minimize the noise from spurious programs, previous works exploit domain-specific knowledge such as utterance groups (Gupta et al., 2021) and abstract programs (Goldman et al., 2018; Wang et al., 2019). Other studies focus on enforcing alignments between relevant utterances and program parts in lexicon level (Dasigi et al., 2019) or phrase level (Wang et al., 2019).

More recently, pre-trained language models have demonstrated outstanding performance on semantic parsing, especially in the table semantic parsing domain, thanks to the large corpus of tables and surrounding natural language utterances (Yin et al., 2020; Yu et al., 2021). These models are trained with tasks devised for natural language and table understanding with the enormous amount of tables, which may not be available when we construct a semantic parser in more scarce domains other than table semantic parsing.

Unlike these approaches, our method can be applied to existing semantic parsers with minimal domain-specific engineering and a relatively small amount of data.

### 7.2 Identifying Programs with Execution Results

Recently, there has been a growing interest in using the execution result of program to guide the training and inference of a deep learning model. Odena and Sutton (2020) introduce a notion of *property signature*, which represents a hypothetical program specified by given input-output pairs. The main difference between property signature and our representation scheme when representing a program is that the former uses a set of simpler programs, and the latter uses a set of related worlds from other examples.

In natural language to code translation, there are attempts to leverage execution results to cluster syntactically different but semantically identical programs and submit the program in the largest cluster for evaluation (Li et al., 2022; Shi et al., 2022). These methods are somewhat similar to soft vote without weight term in our approach, while their goal is to pick the best program at the inference stage rather than filtering spurious programs.

In recent times, the use of self-consistent Chain-of-Thought (CoT) prompting (Wang et al., 2023) has greatly enhanced the reasoning capabilities of large language models. This self-consistency is achieved by decoding multiple reasoning paths and selecting the one with the highest vote score.

Program execution results can be utilized for various purposes. Zhong et al. (2020) propose *test*

*suite accuracy* based on the programs' execution results on tables which are constructed to be likely to distinguish false programs from the gold program. Pasupat and Liang (2016) construct a set of "fictitious worlds" such that the denotations of those worlds are most effective in filtering out spurious programs when annotated by humans. Here, the effectiveness of the world is approximated using the execution results of the programs.

## 8 Conclusion

We proposed a domain-agnostic approach to filter out spurious programs in weakly supervised semantic parsing based on execution results and majority vote. Our assumption was spurious programs are outliers in terms of meaning; thus, we introduced a representation scheme that captures the semantics of programs. Based on these representations, we ran the majority vote to identify and exclude spurious programs from the pool. Our approach showed significant improvements over base models on NLVR and WTQ test set performance, also reporting a new state-of-the-art on NLVR with less domain-specific knowledge than the previous best model.

## Limitations

One limitation of our approach is that it does not help when the program pool is too small. Also, the weight term $W(\cdot)$ played a significant role in achieving state-of-the-art performance. The applicability of our method on domains without any metric of alignment is somewhat questionable, although the use of such metrics (e.g. lexicon coverage) is quite widespread in weakly supervised semantic parsing fields and the construction of the metric has to be done only once for a particular domain. Improving its robustness is one possible future work direction.

## Acknowledgements

This work was supported by National Research Foundation of Korea (NRF) grant funded by the Korea government (No. 2021R1A2C2008855) and Institute of Information & communications Technology Planning & Evaluation(IITP) grant funded by the Korea government(MSIT) [No. 2022-0-00184, Development and Study of AI Technologies to Inexpensively Conform to Evolving Policy on Ethics & NO.2021-0-01343, Artificial Intelligence Graduate School Program (Seoul National University)].

K. Jung is with Automation and Systems Research Institute (ASRI), Seoul National University.

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

# A  Experimental Details

All the experiments were performed using one Nvidia RTX2080ti GPU.

**NLVR**  We used the same model architecture and hyperparameters as Gupta et al. (2021), except for the beam search described in section 5. Regarding the number of worlds $n$, we retrieved top-20 utterances, so $n = 80$ in most cases. We reported the accuracy and consistency on the hidden test set of LLD in Table 1 based on our own experiment because Gupta et al. (2021) do not provide these values. We calculated BLEU score in section 3 with nltk sentence bleu function. A full training process took 20 to 30 hours depending on the settings and random seeds.

**WTQ**  Before we fine-tuned the WTQ base parser (Wang et al., 2019) with our filtering mechanism, it was trained for 15 epochs without any modification. The best model with the highest dev accuracy of 43.2 achieved a test accuracy of 44.4. We fine-tuned this model with 4 different random seeds and reported the average in table 2. Note that the official dev/test accuracy reported in the paper is 43.7/44.5. The program filtering process took 6 to 7 hours and was only done once.

## B  Table Ranking and Filtering for WTQ

Here, we suggest a way to rank and filter the tables since some tables have fewer names than the program pool $Z$, thus less capable or infeasible to support the column and entity replacement process described in section 4.2.

To enhance the feasibility of replacement, we rank the tables according to their likelihood of supporting the replacement process. First, we sort the tables by a score $S_{table} = |S \cap T|/|S|$ where $S$ and $T$ are the multiset[5] of the source table's column types and the target table's column types, respectively. Intuitively, the $S_{table}$ assigns a high score if the target table has more columns than the source table for each column type, prioritizing big tables with various column types that are more likely to facilitate the replacement.

Furthermore, we exclude non-qualifying tables that have strictly fewer column types than the program pool $Z$. Given a table $w$, we first find all occurrences of column and entity names in $Z$ and $w$. Then we construct two dictionaries for type counting: $C_Z$ and $C_w$, whose keys are the column/entity types, e.g., string, number, etc, and values are the number of names of that type in $Z$ and $w$, respectively. Finally, we compare the $C_Z$ and $C_w$'s of training set tables and exclude all the tables with fewer names than $Z$ in any of the types. For example, if $C_Z = $ {string:3, number:2}, a table with $C_w = $ {string:4, number:2} remains but $C_w = $ {string:2, number:4} is excluded. This filtering process ensures that all the names in $Z$ can have a unique name of the same type in the target table.

Additionally, we do not utilize the tables with blank cells as target tables because the programs executed on such tables tend to return errors in high frequency. The program representation is constructed by sequentially executing programs $Z$ on these ranked and filtered tables until the number of worlds used hits $n$ ($n = 40$).

---

[5]Here, the multiset, or bag, is a generalized notion of the set which allows multiple elements with the same value. For example, if the source table has 2 columns with type string, $S$ would have two string elements.