# OpenReview forum: "Weakly Supervised Semantic Parsing with Execution-based Spurious Program Filtering"
_EMNLP/2023/Conference — EMNLP 2023 Main_

### Official Review · Reviewer_ddAc · 2023-07-25

**Soundness:** 4

**Excitement:**

3: Ambivalent: It has merits (e.g., it reports state-of-the-art results, the idea is nice), but there are key weaknesses (e.g., it describes incremental work), and it can significantly benefit from another round of revision. However, I won't object to accepting it if my co-reviewers champion it.

**Paper Topic And Main Contributions:**

The paper tries to address the "ruling out spurious programs" problem in weakly supervised semantic parsing. During the stage of generating several different candidate semantic parses to further the training, the author proposes to cluster the proposed semantic parses based on their "denotation similarity", as approximated by how often two semantic parses output the same results on a set of inputs. They show that the approach is effective on WTQ and NLVR.

**Questions For The Authors:**

A. Can you report some correlation statistics between (1) a program's similarity to the majority vote, and (2) whether a program is spurious? I would love to know the pearson-r correlation, AUC-ROC score, and the mean and std of the similarity scores when a program is spurious/not spurious.

B. This is based on how I understand your algorithm: effectively the proposed algorithm is defining a metric space in the programs, where the pair-wise similarity is how often they agree. What if you filter out all programs that has an average distance too far away from the rest, instead of first computing the majority vote and then filter out those programs that disagree with the vote? (this is just my own curiosity, would not influence the score, though it'd be a plus if the performance actually improves)

**Reasons To Accept:**

- The approach seems interesting.
- The paper is well-written

**Reasons To Reject:**

- The gain in table 1 and 2 feels very incremental to me. The complexity involved in implementing this method might outweigh the empirical benefit. The problem exacerbates as the voting procedure introduces many other hyper-parameters to tune.
- The paper adds little novelty to Shi et al. 2022 (this might not be a fair reason to reject though, given that this paper might have been written before 2022 since all baselines were pre-2020)
- I am unsure how well this intuition holds empirically and can extend to more complex semantic parsing tasks: "programs whose execution results largely deviate from those of other programs in the pool are likely to be spurious, thus filtering them out would improve the training of weakly supervised semantic parsers." I'd prefer having more statistics and results to back this assumption (which the algorithm is based on). See questions below.

**Reproducibility:**

4: Could mostly reproduce the results, but there may be some variation because of sample variance or minor variations in their interpretation of the protocol or method.

**Reviewer Confidence:**

4: Quite sure. I tried to check the important points carefully. It's unlikely, though conceivable, that I missed something that should affect my ratings.

---

> ### Author Rebuttal · Authors · 2023-08-29
>
> We greatly appreciate the reviewer's valuable feedback. In response, we'd like to address several points raised.
>
> Analyzing the Intuition:
>
> In order to address the reviewer's concerns, we've included additional statistics as requested. The Pearson-r correlation between the inverted program score (1-s_i) and the spuriousness label (0 for correct, 1 for spurious) is 0.358. The ROC AUC stands at 0.738. Additionally, the mean and standard deviation of scores (s_i) for correct programs are 0.997 and 0.029 respectively, while for spurious programs, they are 0.899 and 0.155.
>
> Regarding Question B:
>
> We want to clarify that the mechanism you described aligns precisely with our soft vote approach. This approach demonstrates enhancements in WTQ, yet it leads to a decline in performance in NLVR.

---

### Official Review · Reviewer_jkaF · 2023-08-03

**Soundness:** 3

**Excitement:**

3: Ambivalent: It has merits (e.g., it reports state-of-the-art results, the idea is nice), but there are key weaknesses (e.g., it describes incremental work), and it can significantly benefit from another round of revision. However, I won't object to accepting it if my co-reviewers champion it.

**Paper Topic And Main Contributions:**

The paper addresses the problem of spurious correlations in semantic parsing: incorrect programs may coincidentally return the correct answer when they are executed. To address this problem, the paper proposes a method called Execution-Based Filtering. The method relies on the intuition that even though a semantically incorrect program may coincidentally return the correct answer once, it is unlikely to consistently return the same answer as semantically correct programs when run multiple times over other configurations.

The paper evaluates its method on two datasets: Natural Language Visual Reasoning (NLVR) and WikiTableQuestions (WTQ) and shows improvements after it is used to augment existing methods such as Logical Language Design.


**Reasons To Accept:**

The paper proposes a simple method, Execution-Based Filtering, that can augment existing methods such as Logical Language Design to filter candidate programs. The paper shows that on NLVR, Execution-Based Filtering improves accuracy and consistency, and that on WTQ, Execution-Based Filtering improves accuracy.


**Reasons To Reject:**

It would be helpful to see some additional comparisons with methods mentioned in the Related Work section, such as ones using pre-trained language models, self-consistent chain-of-thought prompting, etc. The paper mentions these as relatively recent advancements, so it would be helpful to understand the relationship between Execution-Based Filtering and these methods, e.g. how does the raw accuracy compare, could Execution-Based Filtering be applied to these methods, is there a theoretical relationship, etc.


**Reproducibility:**

4: Could mostly reproduce the results, but there may be some variation because of sample variance or minor variations in their interpretation of the protocol or method.

**Reviewer Confidence:**

2: Willing to defend my evaluation, but it is fairly likely that I missed some details, didn't understand some central points, or can't be sure about the novelty of the work.

---

> ### Author Rebuttal · Authors · 2023-08-29
>
> Thank you for your insightful review. As you pointed out, there are some related works that are applicable to the tasks we addressed in our paper.
> In terms of utilizing PLMs, we believe that transitioning from the LSTM-based models employed in our study to PLMs is a straightforward process. This involves replacing the LSTM seq2seq with a pretrained seq2seq model. It is important to note that most PLMs demand substantial computational resources for effective fine-tuning. Given this consideration, we chose to assess our approach using relatively compact LSTM-based models.
> In addition, the original intention behind Self-consistent CoT did not encompass weakly supervised semantic parsing. Instead, it employed a majority vote mechanism to determine the most likely reasoning path in natural language reasoning tasks. Although our paper acknowledges Self-consistent CoT in the related work section due to its utilization of the majority vote, the primary focus diverges.

---

### Official Review · Reviewer_E67k · 2023-08-06

**Typos Grammar Style And Presentation Improvements:** 1. The problem definition described i…
**Soundness:** 3

**Excitement:**

4: Strong: This paper deepens the understanding of some phenomenon or lowers the barriers to an existing research direction.

**Missing References:**

Looks good to me.

**Paper Topic And Main Contributions:**

This paper presents a domain-agnostic approach aimed at addressing the issue of spurious programs in weakly supervised semantic parsing. The core concept revolves around executing generated programs on compatible worlds and utilizing a voting mechanism to identify and remove spurious examples.

The hard voting scheme involves obtaining centroids for each execution result in every world and calculating a score that reflects the distance from these centroids. By setting a hyper-parameter "t" as a threshold, programs with lower scores are considered more likely to be spurious. In addition, the authors introduce a soft-voting mechanism that takes into account the proportion of denotations.

The experimental evaluation conducted on the NLVR and WTQ datasets demonstrates the efficacy of their filtering mechanism when applied to state-of-the-art weakly supervised parsing methods for both datasets.

**Questions For The Authors:**

1. How are r_i vectors encoded for different result types such as strings, numbers, and boolean?

2. It is not clear how entity replacement works (described in section 4.2)?

3. At t=0.8, 60% of the spurious programs remain in the program pool -> why at t=1.0, do we see too many false positives? This part could be made clearer.

4. Is the code for this work going to be made public?

**Reasons To Accept:**

1. The paper clearly demonstrates the usefulness of their approach for the NLVR and WTQ datasets.

2. The idea to use a voting mechanism based on the execution of programs on various worlds is quite useful.

**Reasons To Reject:**

1. Table 5, is an issue for me. At t=1.0, the f1-score is highest in eliminating the spurious programs, yet it doesn't help in increasing the overall accuracy. at t=0.8, with spurious programs, the gains are higher. Is there an analysis of the kinds of spurious programs that are worse than others? Are there other confounding factors that are overlooked causing this phenomenon? A detailed explanation on this is warranted.

2. I feel that the notations/presentations could be made clearer so as to easily follow the work.

**Reproducibility:**

4: Could mostly reproduce the results, but there may be some variation because of sample variance or minor variations in their interpretation of the protocol or method.

**Reviewer Confidence:**

4: Quite sure. I tried to check the important points carefully. It's unlikely, though conceivable, that I missed something that should affect my ratings.

---

> ### Author Rebuttal · Authors · 2023-08-29
>
> We appreciate the reviewer for providing valuable feedback. Below, we address some of the concerns raised in your comments.
>
> Regarding the Results in Table 5:
>
> We acknowledge that the result presented in Table 5 might appear to be a concern. It is important to note that the result in Table 5 represents the outcome of the final search step within our iterative search process. During this last step, the programs within the beam are mostly accurate. This high accuracy leads to a high optimal threshold for identifying spurious programs (tau=1.0). However, it's worth considering that earlier in the iterative search process, the parser's accuracy is not as accurate as in the last step. This circumstance favors a lower threshold, which in turn facilitates exploration.
>
> Notations and Presentations:
>
> Thank you for pointing out the potential clutter within our notations. We will reflect them if the paper is accepted.
>
> Addressing Questions:
>
> 1. The entries of r_i vectors of different types are stored as they are. This is posible because in the majority vote, we only compare (same/not same) the values in the same position within the r_i vectors.
> 2. To help understanding the entity replacement process, Figure 3 is offered. This illustration demonstrates how we substitute entities occurring multiple times within the program pool (e.g., "column:Wins") with a consistent entity (e.g., "column:Silver"). The new entity is chosen randomly from the set of target table entities.
> 3. In Table 5, the precision reaches 99.4% with a threshold of tau=1.0. This translates to low rate of false positives (0.6%).
> 4. If the paper is accepted, we will make the code publicly available.

---

### Meta-Review · Area_Chair_1hJS · 2023-09-20

**Recommendation:** 4

**Metareview:**

The paper addresses the problem of spurious correlations in weakly supervised semantic parsing with a filtering approach based on "denotation similarity" approximated by how often two programs output the same results on a set of inputs.

Two out of three reviewers acknowledged the rebuttal.

All reviewers value the simple but effective method (on NLVR, Execution-Based Filtering improves accuracy and consistency, and on WTQ it improves accuracy), and give relatively high soundness and excitement scores (3.3 on average for each).  Still, the shortcomings mentioned by reviewers are quite serious, such the fear that the complexity may outweigh the benefits, and that the method may not extend well to more complex tasks, as well as the need for additional comparisons with methods using pre-trained language models, self-consistent chain-of-thought prompting etc. The latter of these is partly addressed in the rebuttal.

---

### Decision · Program_Chairs · 2023-10-07

**Decision:**

Accept-Main

**Comment:**

The paper addresses the problem of spurious correlations in weakly supervised semantic parsing with a filtering approach based on "denotation similarity" approximated by how often two programs output the same results on a set of inputs.

Two out of three reviewers acknowledged the rebuttal.

All reviewers value the simple but effective method (on NLVR, Execution-Based Filtering improves accuracy and consistency, and on WTQ it improves accuracy), and give relatively high soundness and excitement scores (3.3 on average for each).  Still, the shortcomings mentioned by reviewers are quite serious, such the fear that the complexity may outweigh the benefits, and that the method may not extend well to more complex tasks, as well as the need for additional comparisons with methods using pre-trained language models, self-consistent chain-of-thought prompting etc. The latter of these is partly addressed in the rebuttal.